

# Parallel use of threshold parameter variation for tropical cyclone tracking

Bernhard Enz[1,*], Jan Engelmann[1,2,*], and Ulrike Lohmann[1]

[1]Institute for Atmosphere and Climate Science, Swiss Federal Institute of Technology, Zurich, Switzerland
[2]Institute of Computational Biology, Helmholtz Center Munich, Munich, Germany
[*]These authors contributed equally to this work.

**Correspondence:** Bernhard Enz (bernhard.enz@env.ethz.ch)

**Abstract.** Assessing the capacity of numerical models to produce viable tropical cyclones, as well as assessing the climatological behavior of simulated tropical cyclones, requires an objective tracking method. These make use of parameter thresholds to determine whether a detected feature, such as a vorticity maximum or a warm core, is sufficiently strong to indicate a tropical cyclone. The choice of parameter thresholds is generally subjective. This study proposes and assesses the parallel use of many
threshold parameter combinations, combining a number of weaker and stronger values. The tracking algorithm succeeds in tracking tropical cyclones within the model data, beginning at their aggregation stage or shortly thereafter, and ending when they interact strongly with extratropical flow and transition into extratropical cyclones, or when their warm core decays. The sensitivity of accumulated cyclone energy to tracking errors is assessed. Tracking errors include faulty initial detection and termination of valid tropical cyclones and systems falsely identified as tropical cyclones. They are found to not significantly
impact the accumulated cyclone energy. The tracking algorithm thus produces an adequate estimate of the accumulated cyclone energy within the underlying data.

## 1 Introduction

Numerical models are a useful tool to further our understanding of weather and climate, and to make predictions thereof. Within these model simulations, certain features, like tropical cyclones (TCs), can be tracked and their behavior analyzed. TCs
are of particular importance, as they pose an immense threat to human life and assets when they make landfall. The damage caused by TCs is likely to increase with an increase in population and wealth in coastal areas (Pielke Jr. et al., 2008), and increased destructiveness (Grinsted et al., 2019) and frequency (Bender et al., 2010) in a warmer climate.

Model simulations, especially when performed with high horizontal resolution, for large domains, and for extended periods of simulated time, produce vast amounts of output data. To analyze certain features, such as the lifetimes, intensities and tracks
of TCs, they must be identified. Manually tracking every TC is not only very cumbersome, but also complicated by subjectivity. Therefore, an automated and objective algorithm is preferable.

Any tracking algorithm implicitly contains a definition of the tracked system, and then searches for instances where this definition is fulfilled. Within the context of TCs, a commonly used baseline is that a TC is a system with a maximum in vorticity collocated with (e.g., (Chauvin et al., 2006)) or in the vicinity of (e.g., (Bengtsson et al., 1995; Zhao et al., 2009)) a





minimum in sea level pressure. The maximum in vorticity can be used without requiring the minimum in sea level pressure (e.g.,Hodges (1999)). However, this alone does not distinguish TCs from other systems that can occur in the region of interest, such as extratropical cyclones. Therefore, the warm core structure of TCs is usually also searched for during the tracking process, which can be done directly or indirectly.

Directly assessing the warm core is done by defining a temperature anomaly, which compares the temperature at the TC

center to that of the environment at a specified altitude or pressure. For example, Bengtsson et al. (1995) require the temperature anomaly at 300 hPa to be larger than that at 850 hPa, while requiring the sum of the temperature anomalies at 300, 500 and 700 hPa to exceed a threshold value. This ensures that the temperature anomaly is stronger at higher altitude, and that it is not too weak. Chauvin et al. (2006) require a strengthening of the temperature anomaly with height, but only that the 700 and 300 hPa temperature anomalies exceed a threshold value. Zhao et al. (2009) use the mean temperature between 500 and 300 hPa

to define the temperature anomaly, but use the local maximum thereof, which must be within 2° horizontal distance of the sea level pressure minimum.

Indirectly assessing the warm core is done by searching for a pattern that is consistent with the presence of a warm core. For example, Bengtsson et al. (1995) and Chauvin et al. (2006) require the cyclonic wind to weaken with height, which is indirect evidence for a warm core structure. Strachan et al. (2013) require that the vorticity in the TC center is reduced with height,

which is related to the weakening of cyclonic winds with height, and therefore evidence for a warm core. Walsh et al. (2012) combine the direct and indirect criteria, in that they require a positive 300 hPa temperature anomaly and a reduction of wind speed with height.

An alternative method of tracking TCs is provided by Tory et al. (2013), who used the Okubo-Weiss (OW) (Okubo, 1970; Weiss, 1991) parameter and absolute vorticity to form the OWZ parameter. The OWZ parameter reflects the solid body com-

ponent of absolute vorticity, and can thus be used to identify vorticity rich quasi-closed circulations. It is argued that every TC precursor shows increased OWZ, and thus the OWZ parameter is particularly useful to detect the early stages of a TC. Combined with vertical wind shear and relative humidity criteria, the OWZ parameter can then be used to track TCs (Tory et al., 2013; Bell et al., 2018).

Many tracking criteria across many algorithms require that a system exceeds a corresponding threshold value. For example,

the warm core temperature anomaly must exceed the environmental temperature by a predetermined value, or the central vorticity maximum must exceed a predetermined minimum value. This inherently makes tracking algorithms sensitive to the choice of these threshold values (Horn et al., 2014). If the threshold values are too weak, then false positives may be found, and the algorithm cannot be trusted to detect only TCs. If the threshold values are too strict, then TCs that exist in the model can be truncated in the early and late stages, or missed entirely. Furthermore, the horizontal resolution of the model may affect

how sensible a given threshold value choice is (Walsh et al., 2007).

To make the tracking process less sensitive to the choice of threshold values, it is possible to vary these values. An example of this is provided by Camargo and Zebiak (2002), who use more strict threshold values to first identify TCs, and then use relaxed threshold values forwards and backwards in time when a system is detected. This allows them to detect early and late stages of the TC life cycle with the relaxed values while mitigating the pitfall of falsely tracking non-TC systems.





The relative weakness of the defining TC characteristics at early and late stages in the TC life cycle is not the only complication to their tracking, as the existential question of when a TC begins and when it ends can also be asked outside of the scope of tracking. In the North Atlantic basin, only about 40% of TCs form in the absence of baroclinic processes, and about 40% of TCs form in a process called tropical transition (TT) (McTaggart-Cowan et al., 2013), where a precursor storm moves over warm water and attains TC characteristics (Davis and Bosart, 2003). Warm water is required as TCs form predominantly over water warmer than 26°C (Palmen, 1948), though TT events with a strong initial lower-level circulation can form at slightly lower sea surface temperatures (McTaggart-Cowan et al., 2015).

Montgomery and Smith (2014) describe the initial intensification of a weak cyclonic precursor system. They discuss how many mesoscale systems of deep convection, which they name vortical hot towers (VHTs), locally stretch vorticity within this precursor system, and how the thus produced cyclonic vorticity anomalies aggregate while the corresponding anticyclonic anomalies move outwards. This process gradually increases the vorticity of the precursor system, and allows it to develop into a mature TC.

The termination of a TC can occur quite rapidly when they move over land. Other than this rather straight-forward termination, there is also the possibility for a TC to develop characteristics of extratropical cyclones in a process called extratropical transition (ETT) (Evans and Hart, 2003). ETT occurs when a TC moves poleward and encounters a strong meridional temperature gradient, which enables it to form fronts and thereby strong radial asymmetry. This loss of symmetry and the increased vertical wind shear associated with horizontal temperature gradients cause the warm core to decay, such that the system develops an ETC structure. ETT occurs for 46% of TCs in the North Atlantic basin, and transitioning systems account for about half of the systems that landfall (Hart and Evans, 2001).

Both the initial and final stages of TCs are therefore not instantaneous, but rather processes that take a finite amount of time to conclude. A tracking algorithm would therefore preferably detect a cyclone at some point during its development, and cease to detect it at some point during its termination, as this would capture the entire TC phase of the cyclone while allowing for some leeway during the phases immediately before and after.

While publications typically contain a description of how TCs are tracked, it is by no means common that they include an assessment of how well the tracking algorithm performs. As this is a fundamental component on which the data analysis builds, this paper is devoted to introducing a newly developed algorithm, and assessing how well it performs. The new algorithm uses varying threshold values, which allows it to contain both lax and strict threshold value combinations, which are then combined to form a final tracking product.

The remainder of this paper is structured as follows. Section 2 describes the data and methods used to produce model TCs which can then be tracked, as well as the tracking algorithm. Section 3 shows that the numerical simulations are capable of producing viable TC-like vortices. Section 4 assesses at which stage model TCs are first tracked, and section 5 assesses at which stage model TCs are last tracked, and why they terminate. Section 6 explores false positives, and how they are caused. Section 7 assesses the impact of tracking errors on the accumulated cyclone energy (ACE) (Bell et al., 2000). Section 8 assesses the sensitivity of the tracking process to the allowed translational velocity of TCs. Section 9 summarizes the results, and provides the drawn conclusions and an outlook.



## 2 Data and methods

### 2.1 Numerical simulations

ICON version 2.6.1 (Zängl et al., 2015) is used in limited area mode (ICON-LAM) to produce simulation data with which the tracking algorithm can be validated. The simulation domain spans from 120°W to 15°W and from the equator to 70°N. A horizontal resolution of R03B07 is used, which corresponds to a grid spacing of about 13 km. 50 vertical levels are used, with the distance between levels increasing with altitude. The first level is at about 10 meters above the surface, and the model top is at 23 km. A time step of 100 s is used. Shallow and deep convection parametrizations are used (Bechtold et al., 2008). An ensemble of 20 members spanning the entire North Atlantic hurricane season, which runs from the beginning of June to the end of November, is generated for the 2013 season.

ERA5 data (Hersbach et al., 2020) are used to construct the initial state of the simulations, and to prescribe monthly mean values for sea surface temperature and sea ice, and as lateral boundary conditions in 6-hourly intervals. Sea surface temperature, sea ice and boundary conditions are interpolated to individual time steps throughout the simulation. The physical fields that are prescribed at the boundary are zonal, meridional and vertical wind, the logarithm of sea level pressure, temperature, specific humidity, cloud liquid water content, cloud ice water content, rain water content, snow water content and surface geopotential. The first member of the ensemble is initialized at 00:00 UTC on 1 May 2013. The following 19 members have their initial times shifted by 24 hours for each additional member, such that the final member is initialized on 00:00 UTC on 20 May 2013.

### 2.2 Tropical cyclone tracking and evaluation

The tracking algorithm is based on that of Kleppek et al. (2008), which has previously been adapted to identify TCs in ECHAM output data. New features of the presented algorithm are the inclusion of a warm core criterion, and parallelization and threshold variation to address the threshold choice issue mentioned in section 1.

The tracking algorithm requires mean sea level pressure, the vertical component of relative vorticity, and temperature on the 300 hPa isobaric surface on a regular longitude-latitude grid. For the purposes of this study, the chosen resolution is 0.125° x 0.125°, corresponding to about 14 km at the equator, to which the ICON output is remapped.

Initially, all points on the horizontal grid are potential centers of a TC, and the algorithm then excludes all points that do not meet the criteria mentioned below. All points that remain are considered to be TC centers at this stage. This is done for each time step individually, such that no tracks are constructed at this stage. The following criteria need to be fulfilled for a point to qualify as a potential TC center, with the used values listed in Table 1.:

1. The sea level pressure must exhibit a local minimum within a given distance ($p_{s,dis}$)

2. The vertical component of relative vorticity must exceed a threshold value within the lower troposphere ($\zeta_{min}$)

3. The 300 hPa temperature directly above the sea level pressure minimum must exceed the mean 300 hPa temperature within a given distance ($T_{dis}$) by a certain value ($\Delta T_{core}$)



| Variable | | Threshold Values | | | | |
|---|---|---|---|---|---|---|
| $p_{s,dis}$ | [km] | 50 | 100 | 150 | | |
| $\zeta_{min}$ | [s$^{-1}$] | $10^{-6}$ | $10^{-5}$ | | | |
| $\Delta T_{core}$ | [K] | 0.5 | 0.75 | 1 | 1.25 | 1.5 |
| $T_{dis}$ | [km] | 50 | 100 | 200 | 300 | 400 |

**Table 1.** Threshold parameter values used in tropical cyclone tracking.

All thresholds of this list are varied (see Table 1. This is done by prescribing not a single, but multiple threshold values, and all threshold combinations then being used in parallel. This results in multiple distinct sets of identified TC centers, which show considerable overlap especially for strong TCs. Combinations with weak constraints identify weaker TCs more readily, while combinations with strong constraints identify only the stronger phases of TCs. This means that the tail ends of TCs are tracked
by the weak constraints, while the strong constraints are less susceptible to falsely tracked points.

After TC centers at individual time steps are identified, they are merged into tracks. To determine whether two TC centers at consecutive detection steps represent the same system, it is assumed that a TC can have a translational velocity of at most 20 ms$^{-1}$. The sensitivity to this velocity is explored in more detail in section 8. If the two TC centers are within a distance that is consistent with this assumption, they are deemed to be the same TC. Tracks are only retained if they reach a minimum life
time ($\tau$). Within this study, $\tau$ is always 18 hours, meaning that a TC track must endure for at least four consecutive detection steps.

This procedure results in a set of tracks for every parameter combination. These are then merged to form one final set of tracks. To achieve this, every set is searched for instances of the same underlying TC, which typically has a variable track length as stronger constraints in the parameter thresholds produce shorter tracks than weaker constraints. Since the tracking
algorithm aims to include weaker phases, the full length of these tracks is retained. To exclude probable false positives, the number of parameter combinations that identified a given TC is considered. Tropical depressions (TDs, see Table 2) are very weak systems, and are not easily identified. Thus, if 10% of all combinations identify a TD, it is retained. Tropical storms (TS) are more intense, but still weak compared to hurricanes. They are retained if at least 20% of all combinations identify the TC. Hurricanes are rather intense TCs, and are thus comparatively easy to identify. They are retained if at least 50% of all parameter
combinations identify them. These values are subjectively chosen, based on visual inspection of azimuthally averaged wind and temperature fields of a subset of the considered TCs. While this introduces a fixed threshold again, the threshold value issue is reduced to one parameter, and this parameter does not describe the physical properties that the tracked system must exhibit.

As an important purpose of tracking TCs is to determine the activity within a season, the TC activity is quantified by the
accumulated cyclone energy (ACE) (Bell et al., 2000). It is calculated as the sum of the squared maximum wind speeds of all





| Category | Maximum Wind Speed [ms$^{-1}$] | | | | |
|---|---|---|---|---|---|
| TD | | | $v_{max}$ | < | 17 |
| TS | 17 | ≤ | $v_{max}$ | < | 33 |
| Cat 1 | 33 | ≤ | $v_{max}$ | < | 43 |
| Cat 2 | 43 | ≤ | $v_{max}$ | < | 50 |
| Cat 3 | 50 | ≤ | $v_{max}$ | < | 58 |
| Cat 4 | 58 | ≤ | $v_{max}$ | < | 70 |
| Cat 5 | 70 | ≤ | $v_{max}$ | | |

**Table 2.** Saffir-Simpson Hurricane Wind Scale, where TD is a tropical depression, TS is a tropical storm, Cat 1-5 are hurricane categories 1-5, and $v_{max}$ is the maximum instantaneous wind speed.

TCs at either the TS or hurricane stage at 6-hourly intervals, i.e.

$$ACE = \sum_{i=1}^{k} v_{max,i}^2 \tag{1}$$

where $v_{max}$ is the maximum wind speed of a TC at time i, which is documented every 6 hours until the end of the season (k). References to TC intensity follow the Saffir-Simpson hurricane wind scale Saffir (1973), slightly adapted to be consistent with commonly used modern values as seen in table 2.

The first and last detection steps of individual TCs are separated into a number of categories, which are described in table 3. These categories aid in evaluating how early TCs are tracked, and what causes them to terminate.

## 3   Tropical cyclones in the simulation data

Validation of a TC tracking algorithm requires that the model producing the underlying data can represent viable TCs, at least to the extent that the features used in tracking are truly features of the simulated TC. Figure 1 shows the azimuthal mean radial and vertical wind and temperature anomaly of the most intense TC within the dataset. The reference temperatures to determine the temperature anomaly is the azimuthal mean vertical profile at 500 km distance from the center. The third row shows the TC at its highest intensity, and the second and first rows show the TC 24 and 48 hours prior to this, respectively.

The radial wind panels at all three shown times show inflow within the boundary layer, and outflow near the tropopause. This is a well documented feature, which has already been reproduced by very early numerical simulations, where the boundary inflow is recognized as a feature crucial to the TC (e.g., Ooyama (1969)). An expected, though absent from our simulated TC, feature is a shallow region of outflow above the boundary layer (Smith and Montgomery, 2015), as supergradient wind is lifted above the boundary layer and adjusts to gradient wind balance. A further feature of radial wind that is expected following Willoughby (1988), but is absent from our simulated TC, is weak inflow throughout the mid-troposphere, which is linked to





| Category | Occurrence | Description |
|---|---|---|
| **Genesis Categories** | | |
| Single Maximum | 36% | TCs exhibit a single vorticity maximum near the central sea level pressure minimum, or only very weak secondary maxima around a strong central maximum within the first 24 hours. |
| Transitional | 34% | TCs exhibit multiple vorticity maxima near the central sea level pressure minimum, and transition to a single vorticity maximum, possibly with very weak local maxima around a strong central maximum, within 24 hours of first detection. |
| Multiple Maxima | 19% | TCs exhibit multiple vorticity maxima near the central sea level pressure minimum within the first 24 hours. |
| **Termination Categories** | | |
| Warm Core Offset | 52% | The warm core offset relative to the central pressure minimum becomes too large to fulfill the warm core criteria. |
| Translation Velocity | 27% | The translational velocity becomes too large for the algorithm to continue the constructed track. |
| Vanishing Pressure Minimum | 4% | The central pressure minimum vanishes, and thus there is no local minimum to be tracked any longer. |

**Table 3.** Description of genesis and termination categories with the occurrence rate of each category. The total number of tracked TCs is 113, about 12% of tracked TCs are false positives. Not all terminations fall under this categorization.

vortex stretching, and thus TC intensification. Vortex stretching is also linked to changes in vertical velocity with height. All three vertical wind panels show clear updraft regions throughout the vertical extent of the troposphere, beginning at a radius of about 50 km near the top of the boundary layer. The vertical wind speed increases with height in parts of this updraft region, which is indicative of vortex stretching. The panel at 24 hours before maximum intensity in particular shows a deep region of
an increase in vertical velocity with height, which is reversed at a height of around 11 km. This reversal leads to a reduction in vorticity, which manifests itself as a reduction in tangential velocity (not shown), and is collocated with the outflow region. The missing mid-level inflow is thus not indicative of absent vortex stretching, as the vertical wind profile shows clear signs of vortex stretching. The eye of a well developed TC is characterized by subsidence, as shown in Montgomery and Smith (2017) for simulated TCs. While this is present at the time of maximum intensity, it is not present 24 hours earlier, and is
not well developed 48 hours earlier. Possible causes for this are the general weakness of the subsidence, and the small scale of this phenomenon. Further, it has been found that increasing the horizontal resolution of numerical simulations beyond



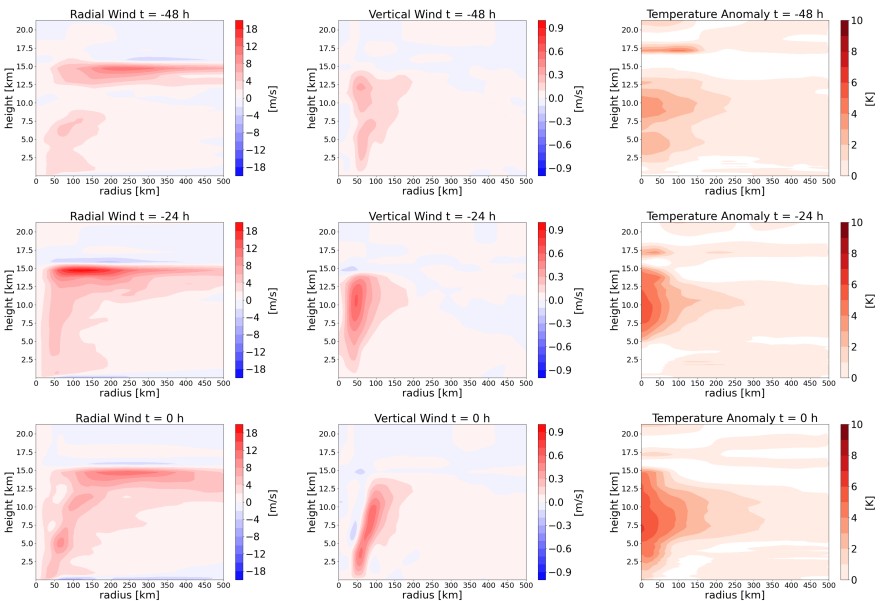

**Figure 1.** Azimuthal mean radial wind (first column), vertical wind (second column) and temperature anomaly (third column) of the most intense TC within the simulation dataset at its highest intensity (third row) and 24 (second row) and 48 (first row) hours prior.

the resolution used in this study can affect the range of downdraft velocities (Gentry and Lackmann, 2010). Generally, the numerical simulations within this study have the capacity to produce the mean secondary circulation features of TCs rather well, even if the more intricate features of secondary inflow are not represented well. Notably, the numerical model can produce

vortex stretching in the lower troposphere, which is relevant to the tracking algorithm as it requires a vorticity maximum.

The temperature anomaly panels for all three time steps show a distinct warm core at the center of the TC. The magnitude of the anomaly increases with increasing TC intensity, which is consistent with the findings of Durden (2013), and the temperature anomaly maxima fall within the height range of 760–250 hPa described therein. The numerical simulations thus have the capacity to produce warm core features that the tracking algorithm requires to distinguish tropical cyclones from extratropical

cyclones.

Figure 2 shows all TCs of a single ensemble member, indicating their category and the percentage of parameter combinations that detected a given track segment. Tracks typically start at very low intensities, and with a lower percentage of threshold parameters detecting the system. The TS and hurricane stages are detected by more parameter combinations, as their structure is more developed. The parameter combinations with weaker constraints are therefore necessary to capture the early stages of

TCs.





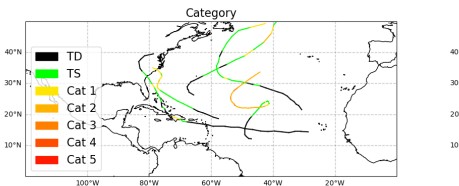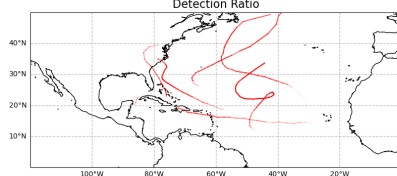

**Figure 2.** Tropical cyclones detected in a single ensemble member. The left panel shows their category, and the right panel shows the percentage of parameter combinations that detected a given track segment. This percentage is used as the opacity of the corresponding track segment.

## 4  Genesis detection

The detection of tropical cyclogenesis is met with a fundamental problem: TCs typically form from a pre-existing disturbance, which gradually develops TC-like characteristics. This means that there is no clear distinction between the pre-existing disturbance and the developed TC. As the tracking algorithm aims to maximize the duration of a TC, how early a TC is detected is sensitive to how weak the most liberal threshold values are chosen. It is thus of interest to investigate how early in the life cycle of a TC the system is detected.

For all detected TCs, the first tracked 24 hours are divided into the three genesis categories listed in Table 3. As the number of TCs is not prohibitively high, this division is done manually to avoid possible oddities in the categorization that an algorithm could produce, though it does introduce some subjectivity. A total of 113 TCs are assessed, of which about 12% are false positives, which are discussed in section 6.

Figure 3 shows a typical example of the single maximum category. This category requires a TC to exhibit a single vorticity maximum near the central sea level pressure minimum, or to have only very weak local vorticity maxima around a strong maximum throughout the entire 24 hour period. About 36% of all tracked systems fall within this category. The horizontal wind speed panels show an asymmetry in the cyclonic wind, which is due to the superposition of the cyclonic wind field and the translational velocity of the TC. Further, the wind speed at the center of the TC is very low, which is a result of the vanishing tangential wind speed towards the center. The TC thus has a developed cyclonic circulation. The vorticity panels, as per the categorization, show a strong central maximum, with comparatively very weak local maxima in the vicinity. The lack of a tracked aggregation phase is not necessarily indicative of a flaw in the tracking algorithm, as TCs can be generated from an extratropical precursor cyclone via tropical transition (Davis and Bosart, 2004), where the precursor cyclone attains TC characteristics. Tropical transition accounts for over a third of cyclogenesis events in the Northern Atlantic (McTaggart-Cowan et al., 2013). The algorithm only tracks these cyclones once the TC characteristics are sufficiently developed. This underlines the importance of the warm core criteria, which serve to distinguish extratropical cyclones from TCs. The temperature anomaly panels show a distinct warm core very close to the center in all three instances. After 24 hours, the warm core is offset to the southeast. While the offset is not immediately relevant to first detection, it shows that the warm core criteria allow for some





offset of the warm core without losing the ability to track the TC. The allowance for this displacement is sensitive to the warm
core threshold parameters, which is reflected in the reduction of the identification percentage for this time step. This in turn
shows that the identification percentage is not sensitive to TC intensity alone. The mean sea level pressure panels serve to show
that the algorithm tracks a genuine low pressure system, and not a spurious local minimum. Notably, the low pressure system
is still tracked when it is embedded in a larger scale pressure gradient, underlining the importance of a parameter that defines
the region within which a point must constitute a local minimum.



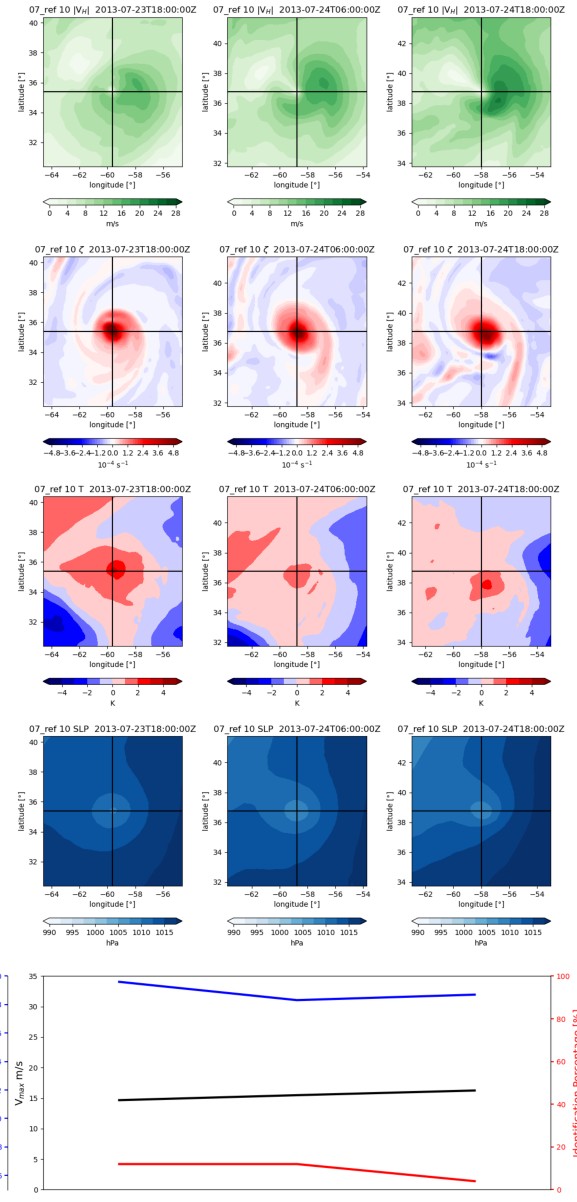

**Figure 3.** Example of a TC in the single maximum genesis category. 850 hPa horizontal wind magnitude (first row), 850 hPa vertical vorticity (second row), 300 hPa temperature anomaly (third row), mean sea level pressure (fourth row) and central pressure (blue), maximum wind speed (black) and identification percentage (red) at the three times shown (fifth row). The first column depicts first detection, the second column shows the TC 12 hours after first detection, the third column shows the TC 24 hours after first detection. Black cross-hairs indicate the TC center.





Figure 4 shows a good example of the transitional category, in that the first time step shows a number of local vorticity maxima of roughly equal magnitude spread throughout a sizeable region. This category requires that at first detection, there are multiple local vorticity maxima in the vicinity of the central mean sea level pressure minimum. Within 24 hours these must give way to a single vorticity maximum, possibly with comparatively very weak local maxima around it (i.e. it must transition into

the pattern that the single maximum category requires from first detection on). This category thus captures TCs that complete an aggregation phase of mesoscale convective systems within the first 24 hours of detection. About 34% of all tracked systems fall within this category. A good example of this category is shown instead of a typical example for two reasons. First, this shows the situation that the variation of parameters aims to track more effectively. Second, the panels 12 hours after first detection show a situation that is reflective of a typical first detection in this category, such that a more typical situation is still

captured by the figure. The wind field panels show a pattern similar to that of the previous category, where cyclonic flow is enhanced in the direction of translation, and reduced in the opposing direction. The location where the tangential velocity is drastically reduced close to the center is slightly offset from the mean sea level pressure minimum, which is typical for the tracked TCs within this category. Once the transition to a single vorticity maximum is completed, this offset typically becomes very small or vanishes entirely. The vorticity panels show many local maxima of comparable intensity at first detection. In

this stage, a key component to the production of vertical relative vorticity is the stretching of pre-existing vertical vorticity. Examination of the vertical wind speeds shows that the vorticity maxima are located where the vertical wind speed increases with height (not shown), which strongly suggests the presence of vortical hot towers, as discussed in Montgomery and Smith (2014), or at the very least strong generation of vertical vorticity through vortex stretching. Therefore, the algorithm appears to be tracking the aggregation stage of mesoscale systems, which then develop into a TC. The algorithm thus successfully fulfills

the goal of capturing this stage for about a third of all tracked systems. However, typically this stage is only detected when the aggregation has progressed substantially, as the panels depicting the TC 12 hours after first detection are more typical for this category. The warm core intensifies throughout the aggregation period, and the location of the maximal temperature anomaly moves closer towards the center of the TC. These two factors cause the identification percentage to increase drastically from the first to the second panel. This shows that the early aggregation phase is within a range where the warm core parameters

are crucial to detection, and the thermal structure of the TC must show some organization. The sea level pressure panels show that the algorithm can track low pressure systems that are rather weak, as is required for capturing the aggregation phase. The maximum wind speed at first detection is close to TS strength in this example, but can be around 10 ms$^{-1}$ in other examples. This low maximum wind speed reflects the early detection in the non-aggregated state. However, the maximum wind speed as tracked by the algorithm does not reflect the maximum wind speed seen in the panels of the first row. This is because for

the maximum wind speed a maximum radius of 100 km is prescribed to not falsely include winds that are not part of the TC circulation. The maximum wind speed of large TCs such as this can thus be underestimated.




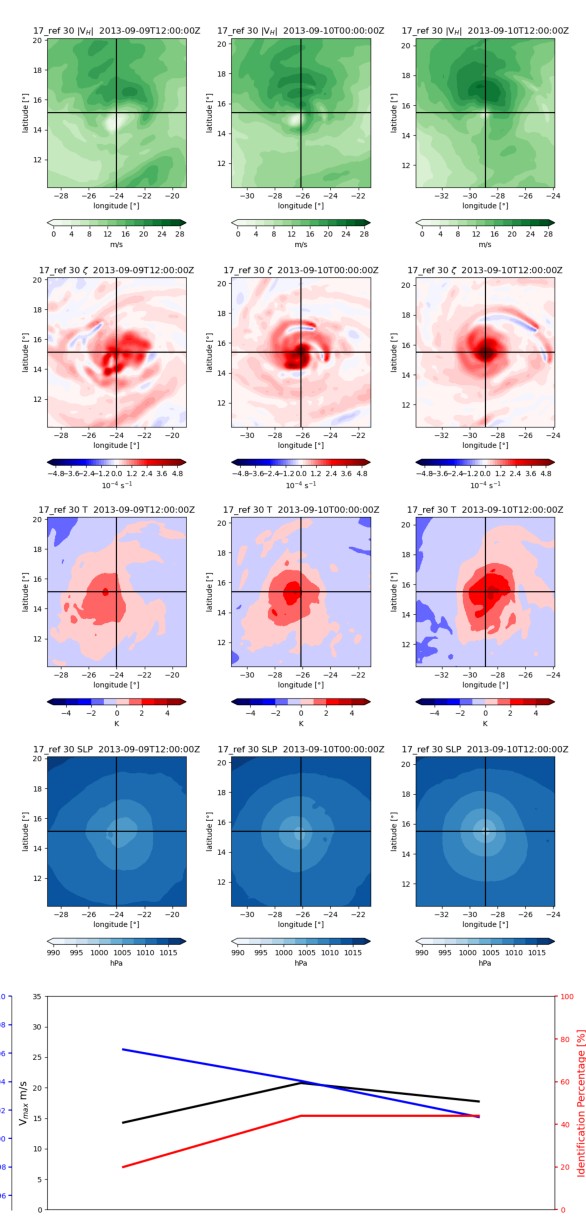

**Figure 4.** As fig. 3 for a TC in the transitional genesis category.





Figure 5 shows a typical example of the multiple maxima category. This category requires that there are multiple vorticity maxima throughout the first 24 hours of the TC, and that there is no singular maximum that is substantially stronger than the others. About 19% of all tracked systems fall into this category. The wind field panels differ substantially from those shown

for the previous two categories. While the very low wind speeds at the center still indicate cyclonic rotation, this is not evident from the winds further away from the center. The cyclonic rotation is thus rather weak, and obscured by environmental winds. This calls into question the usefulness of a maximum wind speed metric, but it will be shown below that these early phases barely impact ACE. This particular example develops into a category 1 hurricane about one week later, which has a much larger impact on ACE. The vorticity panels show only few vorticity maxima, but these increase in number throughout the first

24 hours. This could be due to more VHTs forming, which locally stretch vorticity and aggregate later on. This implies that TCs of this category are detected very early in their life cycle, which is intended. The warm core is barely developed at first, but intensifies throughout the first 24 hours. However, there is no single central maximum, but rather a few local maxima emerge. This unstructured warm core is reflected in the identification percentage, which is barely high enough to not discard this stage of the life cycle. The warm core criteria are thus capable of detecting systems early on, but are not liberal enough to track any

low pressure system with mild diabatic heating. It appears that a substantial region of increased temperature is required for the tracking algorithm to detect a TC, especially when the increased temperature is offset relative to the TC center.



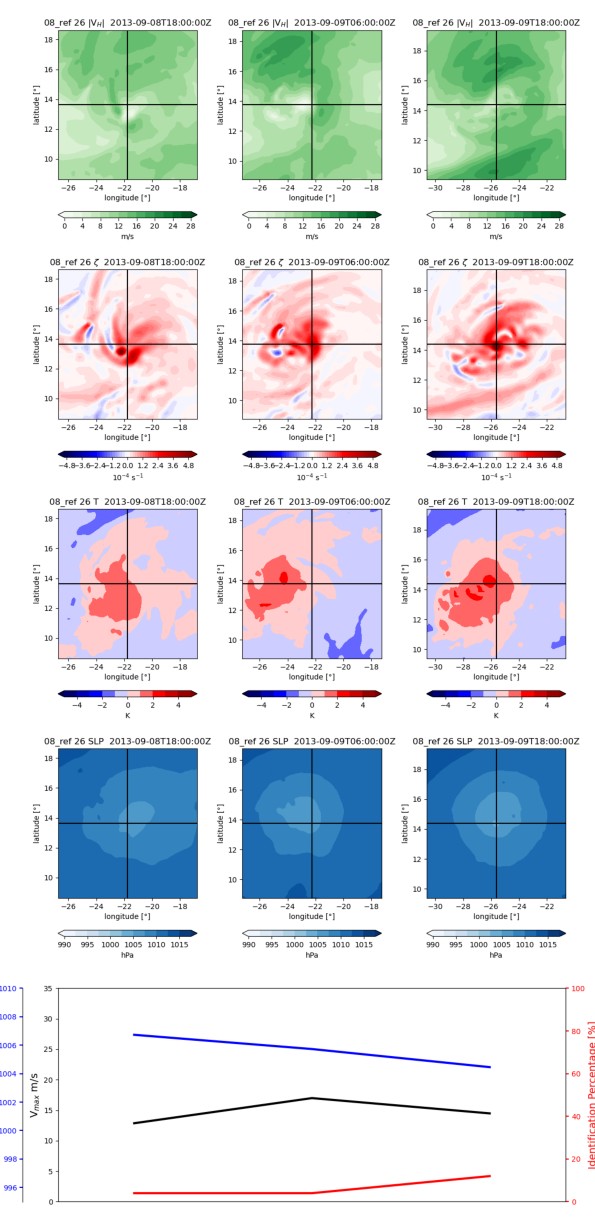

**Figure 5.** As fig. 3 for a TC in the transitional genesis category.





## 5  Tropical cyclone termination

The termination of a TC is, much like genesis, not strictly defined. Therefore, it is investigated what the tracking algorithm deems to be the last time step at which a TC exists, and why it is not tracked further. From a technical standpoint, a system is no
longer tracked when it either is no longer a pressure minimum, when it has very low vorticity, when the warm core criteria are no longer fulfilled, or when it has a very high translational velocity and moves outside of the track construction range within one detection step of 6 hours. Within the used data set, three main causes for TC termination have emerged: (i) The TCs either have a warm core that is offset in a way that increases the environmental temperature such that the warm core criterion is no longer fulfilled, (ii) the warm core weakens substantially, or (iii) the TC moves too fast to be connected to previous detection
steps. Typically, at least two of these processes occur in parallel. The following figures resemble those of the previous section, but now span only the last 6 hours of the TC in the first and second column, and a third column shows plots 6 hours after the TC is last tracked, centered on the final TC position. Mean sea level pressure contours are overlaid on the horizontal wind magnitude, vorticity and temperature anomaly filled contours to better identify where the pressure minimum is located relative to features within these plots. The temperature anomaly within the figure (not during the actual tracking) is calculated using
a $4°x4°$ square centered on the center of the cross-hairs as a reference, as this is somewhat reflective of how the temperature anomaly is calculated by the tracking algorithm.

Figure 6 shows an example of a TC where the tracking algorithm finds a pressure minimum with sufficient vorticity, but where the warm core is offset relative to the pressure minimum, and the warm core is weakening in intensity. The wind magnitude panels show a cyclonic wind field around a pressure minimum even after the TC is no longer tracked, and the
vorticity panels show that there is sufficient vorticity to fulfill the vorticity criterion at all times. In the third column, the distance between the pressure minimum and the last tracked position is also well within the permissible distance that would allow for a track to be constructed. Therefore, the TC must be terminated by the warm core criteria no longer being fulfilled. This is because the weakening warm core is positioned east-south-east of the pressure minimum, and some distance away from it. The pressure minimum is thus located towards the edge of the warm core. This combines a rather weak anomaly above the
pressure minimum with an environmental temperature that is heavily impacted by the presence of the warm core, such that no warm core is detected. The weakening warm core throughout this period is not directly visible in the figure because the reference temperature is progressively reduced, but the absolute temperature of the maximum does indeed decrease, and the area of elevated temperature decreases as well. This aids in offsetting the warm core location from the pressure minimum, as the area of strong temperature anomaly is reduced. This scenario is the most common, with about 52% of cases terminating
due to an offset of the warm core position.

Figure 7 shows an example where the TC has a translational velocity that is too large for a track to be constructed. While between the last two tracked steps the TC moves about $3°$ to the north and $2°$ to the east, the TC then accelerates and moves about $4°$ to the north and $3°$ to the east. This makes it too fast for a track to be constructed. The wind field panels indicate that there is still cyclonic rotation around a pressure minimum, and the vorticity panels show that there is still strong vorticity
associated with the TC. The warm core criteria also seem to be fulfilled. However, a feature that is typical for such cases is that



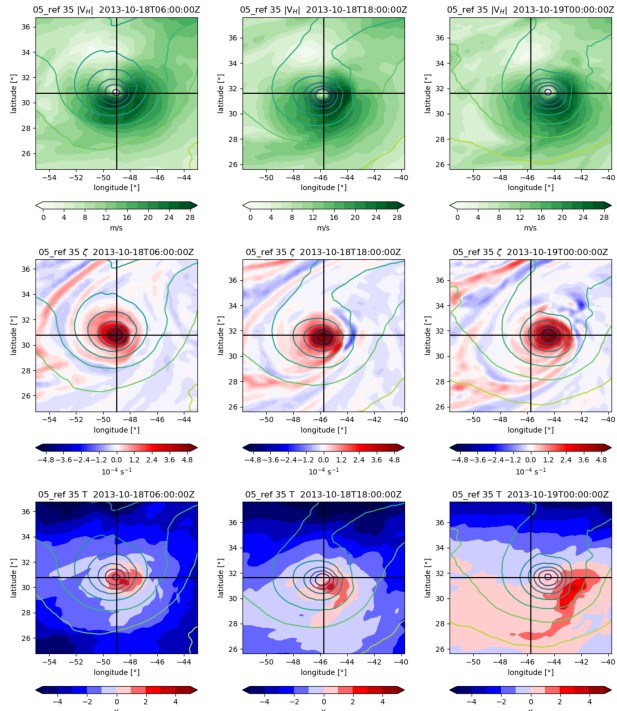

**Figure 6.** Example of a TC in the warm core offset termination category. 850 hPa horizontal wind magnitude (first row), 850 hPa vertical vorticity (second row), 300 hPa temperature anomaly (third row), with overlaid mean sea level pressure contour lines. Black cross-hairs indicate the tracked TC center for the first two columns, and the last tracked TC center (i.e. that of the second column) for the third column.

there is a strong environmental temperature gradient, and cold air enveloping the TC in a cyclonic fashion. This is indicative of extratropical transition (Evans and Hart, 2003), which may also be the cause for the increasing asymmetry in the vorticity field surrounding the TC. The intensity of the warm core is substantially reduced within the shown 12 hour window, which is consistent with the erosion of a warm core during extratropical transition. Therefore, it appears that TCs that have a too high

translational velocity tend to be TCs that are interacting with extratropical flow and are undergoing extratropical transition. While it is reasonable to exclude transitioned TCs, the precise moment where tracking is terminated is not controllable with this algorithm. Further, the tracks do not terminate explicitly because of the transition process, but because the TCs accelerate, making this termination scenario convenient, but accidental. About 27% of TCs terminate due to a too large translational velocity.

Far less common is the vanishing of the sea level pressure minimum. This occurs when a substantially larger low pressure system absorbs the local pressure minimum, which causes the TC to weaken and its pressure minimum to vanish towards the edge of the larger low pressure system. This is the case for 4 TCs within this data set. There is one TC that approaches the eastern boundary of the domain at 15°W, where it weakens and is eventually no longer tracked. Due to the proximity to the boundary, it is not counted in any of the other categories.



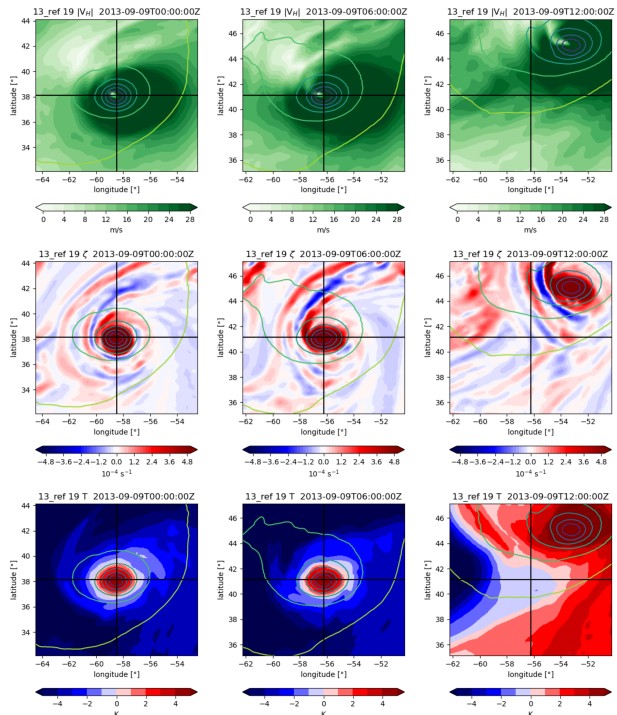

**Figure 7.** As fig. 6, but for a TC in the translational velocity termination category.

In conclusion, tracked TCs reliably terminate due to the erosion or offset of the warm core, due to interaction with extratropical flow, or due to the vanishing local pressure minimum at their center. The translational velocity criterion aids in terminating TCs when they interact with extratropical flow, even when the warm core is still present. While this is convenient, as TCs terminated in this manner show strong asymmetry instead of the typical radial symmetry of a TC, it is not intended and not controllable by parameter choice. The maximum translational velocity parameter serves to form tracks out of individual detection steps, and must be chosen to fulfill that function. It can therefore not be chosen freely and adapted to optimize its function as a termination criterion.

## 6  False positives and other tracking issues

Naturally, there are limitations to the tracking algorithm. Other than beginning to track TCs too early or terminating them too late, there are also instances of tracked systems that cannot feasibly be considered TCs. About 12% of all tracked systems are false positives. Almost all of these cases show traits of extratropical cyclones. Figure 8 shows a typical example of such a case. The wind field at all shown times shows an elongated band of high wind speeds, but no central minimum. There appears to be no discernible center of cyclonic flow near the track center, indicating that the wind field is inconsistent with the existence of a TC at this location. The vorticity fields at all shown times further substantiate this, as elongated bands, partially with





alternating signs, are inconsistent with a developed TC, and also inconsistent with the aggregation of local maxima caused
by vorticity stretching as seen in previous examples. The temperature fields do not show warm cores, but positive anomalies
in an environment with a strong temperature gradient. The presence of strong negative anomalies reduces the environmental
temperature sufficiently for the algorithm to detect what it believes to be a warm core, as the positive anomaly at the center
does not need to be pronounced or confined to a small region to be above the environmental mean temperature. The sea level
pressure fields show that the center of the tracked system is not where the pressure minimum of the low pressure system is
located. Instead, the distance from the true pressure minimum of the system increases with time. It is thus concluded that
the algorithm can mistakenly track frontal structures in extratropical cyclones, as these can show a local pressure minimum,
sufficient vorticity, and a strong temperature gradient that technically fulfills the warm core criterion, even though it is not
a true warm core. Other, atypical cases of false positives are local pressure minima with some positive vorticity that are in
an environment with a strong temperature gradient, which fulfills the warm core criterion even though no true warm core is
present.

Outside of false positives, it is possible for the algorithm to detect a TC, but to falsely identify first detection. Figure 9 shows
a case where first detection is close to the TC's true location. At first detection, the minimum in the wind field that indicates the
center of the cyclonic rotation is some distance away to the northeast, as is the vorticity maximum. As the vorticity maximum
is rather broad, and only a few spurious and comparatively weak local maxima are located outside of it, there appears to have
been an aggregation phase prior to detection. The sea level pressure shows a minimum close to the wind speed minimum and
vorticity maximum to the northeast of the tracked center. In the following detection steps the system is tracked correctly. This
false tracking is likely caused by the pressure minimum to the northeast of the first detected point not fulfilling the warm core
criteria. A local minimum outside of this - the falsely tracked center - fulfills all criteria in what appears to be a single or at most
two parameter combinations, as indicated by the very low identification percentage at this time. This falsely tracks a system
earlier than it otherwise would be at a location that does not reflect where the system is located. It should be noted that this is
the only case within this data set where first detection of a TC is at the wrong location.

Similar to falsely identifying the beginning of a TC, it is possible for the algorithm to detect a TC correctly, but to then
not terminate it early enough. Figure 10 shows a case where a legitimate TC is identified, but the algorithm then detects a
local pressure minimum adjacent to a stronger low pressure system. This local minimum has sufficient vorticity to fulfill the
threshold requirement, and is in a region with a sizeable temperature gradient. This allows the system to fulfill the warm core
criterion without having a warm core, causing the algorithm to continue to track a system beyond extratropical transition. Two
such cases exist within the used data set.

In conclusion, there are features tracked by the algorithm that are not TCs, and TCs are not always tracked correctly. False
positives are rather rare (about 12% of all tracked systems). The impact of these errors on ACE is explored in the following
section.





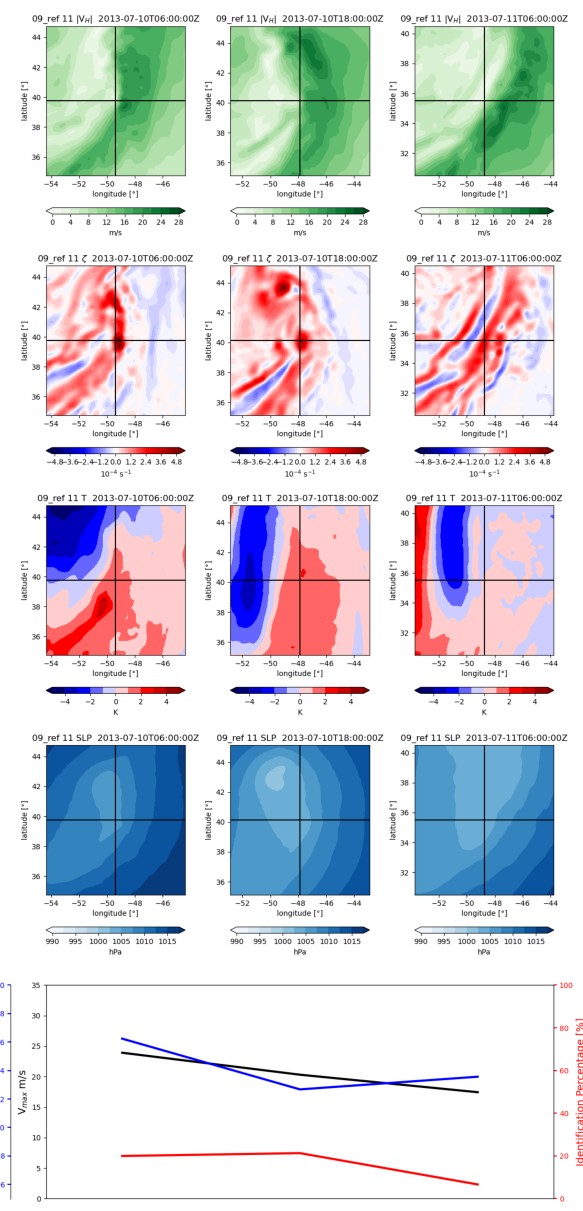

**Figure 8.** As fig. 3, but showing a falsely tracked frontal structure.

## 7  Tracking error impact on ACE

Within this thesis, the main metric used to describe tropical cyclone activity is ACE. To determine the effect of tracking errors on ACE, a few different ACE calculations are considered. Figure 11 shows box plots of the 20 ensemble members comparing the different calculations. The first is full ACE, which includes all TCs at stages of TS strength or higher, including the false



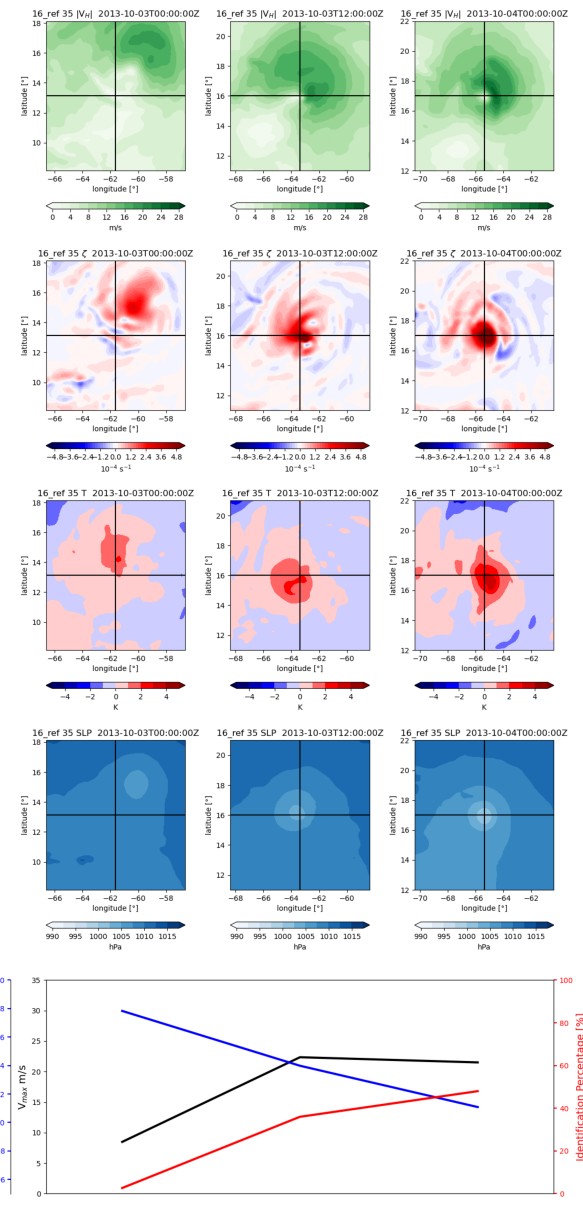

**Figure 9.** As fig. 3, but showing a falsely tracked initial location.

positives. This distribution is used as a reference for the following three. The second calculation is the same, but excludes all manually identified false positives. A one-sided t-test is performed to determine whether the distributions differ significantly, and the resulting p-value is 0.54. Therefore, there is no significant effect on ACE when false positives are included in the calculation. This is likely because the identified false positives typically have short lifetimes, and are not very intense, and therefore do not contribute substantially to ACE.



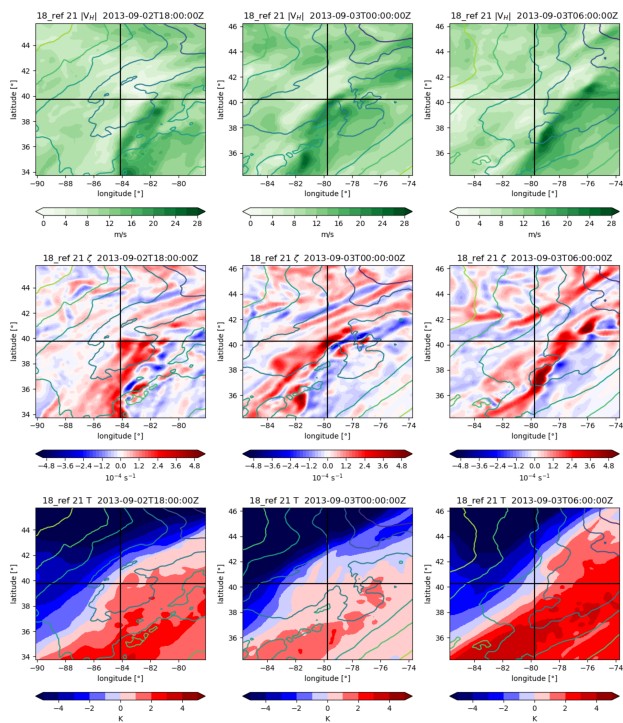

**Figure 10.** As fig. 6, but showing a false continuation of a TC that no longer exists.

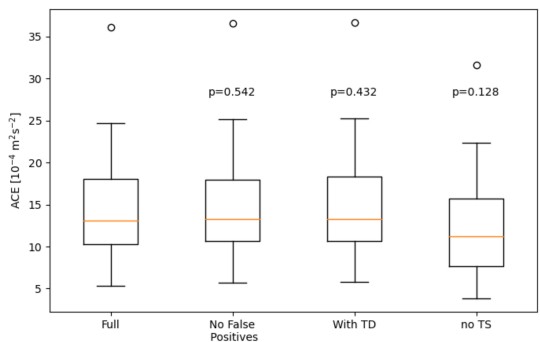

**Figure 11.** Box plots of ACE with the p-values of a one-sided t-test that assesses the difference of the means of various ACE calculations.

The third calculation includes all TCs at stages of TD strength or higher. The underlying rationale is that extending the tail ends of the tracks, and having varying track lengths depending on tracking parameter threshold choices, could impact the energy produced by individual TCs, which would not be captured by the regular ACE calculation. However, a one-sided t-test yields a p-value of 0.43, which shows that no significant difference is produced by the inclusion of the TD stage. The





track extension of the parameter combinations with weak constraints thus allow for very early stages in TC development to be
tracked without significantly affecting the energy produced by individual TCs.

The fourth calculation only includes the hurricane stage of TCs. This is done to show that extending the tail ends of TCs does not significantly impact the ACE contribution of individual TCs. In principle, the algorithm could extend the lifetime of TCs for too long, or could track them too early. As shown previously, this could be an issue when extratropical and tropical transition are involved, as there is no explicit treatment of these processes. Removing all TS stage data from ACE is an estimate
for an upper bound of the impact this could have, which is intentionally chosen to be overestimated. A one-sided t-test yields a p-value of 0.13, meaning that the difference is not significant at the 90% level. Therefore, even if all tracks included TS stage data due to some flaw of the algorithm, ACE would still be adequately represented. This is especially the case when considering that the presented scenario is intentionally chosen to be an upper bound, and that most TS stage data truly reflect a TS in the data.

The impact of flaws in the tracking algorithm on ACE is thus concluded to be negligible, and the extension of the tail ends of tracks does not significantly increase the total energy produced throughout the full life cycle of individual TCs. The tracking algorithm is concluded to be capable of adequately capturing ACE within the underlying data.

## 8  Sensitivity to translational velocity

From the detection of TC termination, it has become apparent that interaction with extratropical flow can cause TCs to speed
up drastically, and that this accidentally, though conveniently, terminates TCs. This may cause the choice of the maximum translational velocity to be more impactful than intended. Therefore, the sensitivity to this threshold is investigated.

The maximum translational velocity threshold used for the preceding analyses is 20 ms$^{-1}$. Figure 12 shows histograms of the translational velocities of HURDAT2 systems of TD, TS and hurricane (HU) categories. This velocity is calculated as the great circle distance between the TC's current location, where the categorization is made, and the location 6 hours prior,
divided by six hours. Non-synoptic times are not considered.

The histograms show that it is extremely rare for observed TCs to move at a mean velocity of 15 m/s or faster. The TC tracking algorithm is thus applied three times, with the maximum allowed translational velocity set as 15, 20 and 25 m/s, respectively. Figure 13 shows the corresponding box plots. There is an increase in ACE with an increase in maximum translational velocity, as a relaxation of this parameter naturally constructs longer tracks. However, with p-values of 0.60 and 0.32
for the 15 m/s and 25 m/s cases, respectively, there is no significant difference in the distribution of ACE. An increase in the maximum translational velocity inherently bears the risk of introducing erroneous tracking, and therefore an increase beyond 20 m/s does not seem necessary or appropriate. A reduction to 15 m/s appears to be feasible, but fig. 12 shows that there are still a few observed TCs with a mean velocity above 15 m/s. Therefore, using 20 m/s appears to be the most appropriate maximum translational velocity.





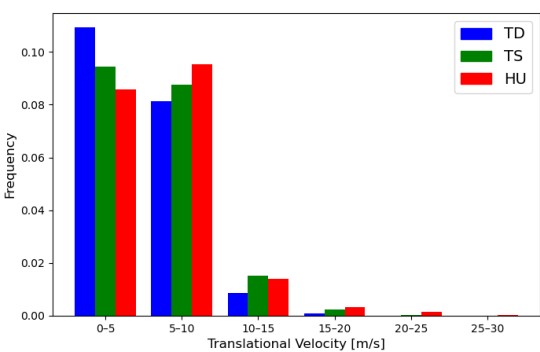

**Figure 12.** Normalized histogram of translational velocity of HURDAT2 systems for tropical depressions (TD, blue), tropical storms (TS, green) and hurricanes (HU, red).

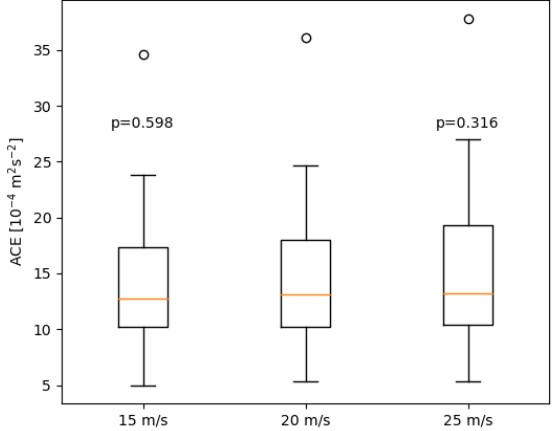

**Figure 13.** Box plots of ACE with the p-values of a one-sided t-test that assesses the difference of the means of ACE using different maximum translational velocities.

## 9 Conclusions

A tracking algorithm for tropical cyclones was developed for use with ICON output data. The algorithm successfully tracks tropical depression, tropical storm and hurricane strength systems. About 36% of TC tracks begin with a strong central vorticity maximum, and about 34% begin with an aggregation of multiple vorticity maxima, in line with the VHT theory of TC cyclogenesis Montgomery and Smith (2014). About 19% of TC tracks begin with an ongoing aggregation process, and remain within this process for at least the first 24 hours. About 12% of tracked systems are false positives.





The benefit of threshold parameter variation is apparent in the tracking of weak systems, especially at early stages of the TC life cycle. In particular, the relaxation of the warm core criterion allows for a larger distance between the warm core temperature anomaly maximum and the sea level pressure minimum. This distance becoming too large is the leading cause of track termination in TCs that are weak, but could feasibly be tracked for longer. However, it also allows for structures

that are not warm cores to be falsely tracked. This is evident from the about 12% false positives, but also in TC termination, where a wrongful continuation of a system can be tracked. As the use of the OWZ parameter is particularly useful in detecting cyclogenesis, a comparison between the threshold parameter variation and the OWZ parameter for tracking early stages of a TC may be in order. Joint use of the two methods could be feasible. Variations in the vorticity threshold do not seem to be of much importance, as TCs that are tracked typically fulfill even the most strict vorticity criterion. A threshold of $10^{-6}$ s$^{-1}$ could

thus be sufficient, as this constrains the resulting tracks to those with positive vorticity.

Specifically for weak TCs, the maximum wind speed is often underestimated. This is because they often have a radius of maximum winds that is larger than the maximum radius within which the algorithm determines maximum wind speeds. An increase in radius may provide more accurate results, though care needs to be given to not accidentally include winds that are not part of the TC. It is possible that using a variable radius threshold based on central pressure could be beneficial, as weak

TCs in particular are affected. However, a more accurate identification of TC genesis should precede this to provide a more solid basis for the exact nature of the radius variability.

The warm core criteria are central to discriminating between TCs and other low pressure systems. They are also responsible for most track terminations. Therefore, these criteria in particular need to be refined. Within the used data set, strong environmental temperature gradients have caused the warm core criteria to be fulfilled even in the absence of a warm core. A possible

solution to this would be to not only use the environmental mean temperature as a reference, but to introduce an additional requirement of having to reach a minimum positive anomaly within every quadrant. This would cause the criterion to not be fulfilled when one side of the identified system is substantially warmer than the other side, i.e. when a strong environmental gradient is present. Further, the offset of the temperature anomaly maximum from the pressure minimum could be treated explicitly. This could consist of introducing a new threshold parameter which determines the maximal allowed distance between

the two extrema, and could be made dependent on the central pressure, as this is particularly relevant for weak systems.

While cyclones could be identified and tracked by a dedicated algorithm for extratropical cyclones after the transition, there is no guarantee that this does not result in gaps between the tropical and extratropical stage of cyclones during the transition. A further possible addition to the algorithm would therefore be an explicit treatment of extratropical transition. Currently, this process is only included via the warm core criteria, which is not only immensely inelegant, but also allows for no control over

how this process is tracked. A clear definition of the transition process is given in Evans and Hart (2003), and could be used within simulation data. This would allow for extratropical phases to be tracked as well, while providing a clear distinction between tropical and extratropical systems.

Furthermore, the effect of tracking errors on ACE has been investigated. The false positives appear to only have a minor impact on ACE. A hypothetical case where all identifications of TD and TS strength systems are false is used to show that

even if this were the case, the impact on ACE would be insignificant. Therefore, the ACE value calculated from the tracking





algorithm output is concluded to reflect the true ACE value within the simulation data. Changes to how cyclogenesis is being tracked can thus be made without substantially impacting ACE. Explicit treatment of the extratropical transition process might have a larger impact, as these systems can still have somewhat high wind speeds, and extratropical transition occurs frequently.

*Code and data availability.* The code of the tracking algorithm is published on Zenodo, DOI:10.5281/zenodo.7331861. The simulation data
are available upon request.

*Author contributions.* Jan Engelmann wrote the tracking algorithm and performed a first validation thereof, which is not part of this study. Bernhard Enz supervised this work, and performed the validation presented here as part of his PhD thesis under the supervision of Ulrike Lohmann.

*Competing interests.* The authors declare that they have no conflict of interest.





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
