# Peer review of "Use of threshold parameter variation for tropical cyclone tracking"

_Geoscientific Model Development, 2022_

## Referee Comment (RC1)

**Review of manuscript titled "*Parallel use of threshold parameter variation for tropical cyclone tracking*" by Enz et al. submitted to GMD**

General comments:

       Using TC-permitting ICON over a limited domain, the authors explored the sensitivity of tracked TCs over the North Atlantic to various combinations of parameters in a tracking algorithm. The initial/genesis and termination stages of TCs have been examined. Generally, valid genesis detections prefer weak constraints from the parameters.

       Overall, I think this paper is well written and the methodology is clearly illustrated. There are several things need to be clarified. Please see my specific comments below.

**Specific comments:**

Title: "Parallel" sounds like parallel computing. How about "Combinations of threshold parameters variation for tropical cyclone tracking"?

Line 17: "increased frequency in a warmer climate": Bender et al. (2010) is cited here. However, I don't think there is enough evidence indicating increased TC frequency in a warmer climate, isn't it?

Line 115: the vertical component of relative vorticity is on 850 hPa?

Line 135: in terms of the minimum lifetime, how many cases have exactly 4 consecutive time steps (and what is the percentage)? Could variations of this parameter make a difference? I would assume that short-lived cases are usually associated with weak intensity.

Line 165 and Figure 1: the boundary layer inflow is a key feature, as the authors mentioned. However, the signals in the boundary layer are pretty weak in terms of Fig. 1 (very light blue shadings). I am wondering if the authors can compute the convergence of wind. For example, Fig. 9 from a recent paper highlights the role of boundary layer inflow by the convergence of the wind field, where the magnitude of convergence also increases with TC intensity. (https://journals.ametsoc.org/view/journals/clim/aop/JCLI-D-22-0199.1/JCLI-D-22-0199.1.xml)

Figure 2: A color bar is needed for Fig 2b.

Line 253-256: I don't understand this part. More explanations are needed.

Line 274-277: rephase these lines

Around line 301 and Figure 7: the tracking algorithm stops when the translational velocity is too large, probably due to the background mean flow. For this case, what is the background mean flow like? E.g., the steam function at 500 hPa or some other variables.

Section 7: in addition to ACE, how about using Power Dissipation Index (PDI)? Do the results stay similar?

Section 9: is it possible that phase-dependent threshold parameters are introduced in a tracking algorithm? Basically, the genesis stage would use a set of parameters that are better for identifying weak TCs, while the termination stage would use a different set of parameters. Perhaps this requires a name tag indicating what the stage is at each time step and thus controlling the choice of parameters online.

---

## Author Comment (AC1)

We would like to thank the three reviewers for providing detailed reviews. They were very helpful in improving the manuscript. We address every point individually below, with the reviewer comments in black and our replies in blue. The reviews are ordered in chronological order.

**Reviewer #1**

Review of manuscript titled "Parallel use of threshold parameter variation for tropical cyclone tracking" by Enz et al. submitted to GMD

General comments:

Using TC-permitting ICON over a limited domain, the authors explored the sensitivity of tracked TCs over the North Atlantic to various combinations of parameters in a tracking algorithm. The initial/genesis and termination stages of TCs have been examined. Generally, valid genesis detections prefer weak constraints from the parameters.

Overall, I think this paper is well written and the methodology is clearly illustrated. There are several things need to be clarified. Please see my specific comments below.

Specific comments:

Title: "Parallel" sounds like parallel computing. How about "Combinations of threshold parameters variation for tropical cyclone tracking"?
While the referenced implementation of the method does use parallel computing, this is not strictly required, so I agree with this point. The new title is "Use of threshold parameter variation for tropical cyclone tracking," as we believe that this reflects how the described method differs from others, without implying a specific implementation of the method.

Line 17: "increased frequency in a warmer climate": Bender et al. (2010) is cited here. However, I don't think there is enough evidence indicating increased TC frequency in a warmer climate, isn't it?
This is a mistake in the manuscript. The correct statement that was intended is that the frequency of category 4 and 5 hurricanes is predicted to increase, which in turn increases the destructiveness according to Grindsted et all. (2019). This is now properly stated in the revised manuscript.

Line 115: the vertical component of relative vorticity is on 850 hPa?
The level at which vertical vorticity is evaluated is now specified. This level is 2.5 km, because ICON internally uses model levels at a constant height above the surface, and therefore a constant geometric height over the ocean. As now argued in the revised manuscript, the core function of the vorticity criterion is to ensure that the system has cyclonic rotation, and therefore the exact vertical position where it is evaluated is not immediately relevant.

Line 135: in terms of the minimum lifetime, how many cases have exactly 4 consecutive time steps (and what is the percentage)? Could variations of this parameter make a difference? I would assume that short-lived cases are usually associated with weak intensity.
There is some sensitivity to the minimum lifetime. However, this is not associated with weak TCs as much as it is associated with short-lived false positives. This is now clearly stated in the revised manuscript.

Line 165 and Figure 1: the boundary layer inflow is a key feature, as the authors mentioned. However, the signals in the boundary layer are pretty weak in terms of Fig. 1 (very light blue shadings). I am wondering if the authors can compute the convergence of wind. For example, Fig. 9 from a recent paper highlights the role of boundary layer inflow by the convergence of the wind field, where the magnitude of convergence also increases with TC intensity.

(https://journals.ametsoc.org/view/journals/clim/aop/JCLI-D-22-0199.1/JCLI-D-22-0199.1.xml)
Figure 1 mainly serves to show that the TCs produced in the simulations show the necessary features for the tracking algorithm to detect them as TCs. We fully agree that the inflow in the boundary layer is rather weak, and this is now clearly stated in the manuscript. Further, the figure has been reworked slightly to make the inflow velocity clearer. We believe that investigating wind convergence in the boundary layer in more detail is not required for the figure to fulfill its purpose. It could detract from the purpose of the manuscript by introducing a tangent that is unrelated to the tracking algorithm.

Figure 2: A color bar is needed for Fig 2b.
A colorbar has been added to the figure.

Line 253-256: I don't understand this part. More explanations are needed.
These lines have been reworded for more clarity in the revised manuscript.

Line 274-277: rephase these lines
The lines have been reworded and should be clearer now.

Around line 301 and Figure 7: the tracking algorithm stops when the translational velocity is too large, probably due to the background mean flow. For this case, what is the background mean flow like? E.g., the steam function at 500 hPa or some other variables.
The 500 hPa streamfunction has been investigated, but no meaningful clarification can really be gained from it. It is therefore not discussed in the revised manuscript.

Section 7: in addition to ACE, how about using Power Dissipation Index (PDI)? Do the results stay similar?
The results using PDI are similar due to the strong similarity between ACE and PDI. Therefore, we believe that adding PDI to the manuscript does not add much value.

Section 9: is it possible that phase-dependent threshold parameters are introduced in a tracking algorithm? Basically, the genesis stage would use a set of parameters that are better for identifying weak TCs, while the termination stage would use a different set of parameters. Perhaps this requires a name tag indicating what the stage is at each time step and thus controlling the choice of parameters online.
While it is possible to use such an approach in general, it is not really supported by how the presented algorithm is structured. The presented algorithm only constructs tracks in a second step, after all potential TC centers have been identified. Therefore, during this identification process, the algorithm has no concept of which potential TC belongs to another from a previous time step. Using different parameters depending on the current stage of a TC would therefore require a complete revision of the algorithm, which is beyond the scope of this paper.

**Reviewer #2**

Review comments on "Parallel use of threshold parameter variation for tropical cyclone tracking" by Enz et al.

Manuscript ID: gmd-2022-279

Recommendation: Accept with minor revisions

General comments:

This manuscript investigated using multiple combinations of different threshold parameters for tracking tropical cyclones from the 20 member ensemble regional ICON-LAM model simulations for the 2013 North Atlantic (NATL) hurricane season. It is shown that using multiple combinations of thresholds is beneficial in tracking/identifying TC genesis early stages as well as the extratropical transition or decaying phases. Overall, the manuscript is well organized and prepared. I only have a few minor concerns (see details in the specific comments below) before the manuscript can be accepted for publication.

Specific comments:

The tracker input data used in this work were from the 20 member ensemble ICON-LAM model simulation for the 2013 NATL hurricane basin/season driven by ERA5 reanalysis data. In the manuscript, it seems to me that, the authors only showed/discussed the tracking results of these 20 member ensemble ICON-LAM simulations, but did not compare the tracking results against the observed (best track) data (or even just the tracking results by using the ERA5 reanalysis data though with relatively low horizontal resolution). I would suggest adding some analyses/comparisons to demonstrate the capability and effectiveness of the ICON-LAM model simulation together with the tracking algorithm used here in terms of capturing/reproducing the TCs for the 2013 NATL hurricane season.

A comparison to observed best track and ERA5 data is omitted intentionally. This is because the simulated 2013 season differs strongly from observations. As only a single season is simulated, even though 20 ensemble members were generated, the underlying data cannot be reflective of the used ICON-LAM configuration's capability to reproduce TC seasons. This is further exacerbated by 2013 being a rather peculiar year in terms of TC activity and its forecasting. The year 2013 was used due to historical reasons, in that the simulations were already performed, and these are the data we had available. We therefore deemed it best to use the data as an arbitrary set of ensemble members that generated TCs, instead of an actual representation of the 2013 season. This was not stated in the manuscript with sufficient clarity, which has now been rectified, and is clearly stated in section 2.1.

Could the authors comment on how the different tracking threshold parameter values are determined? Are they specifically chosen for the ICON-LAM regional model configurations (e.g., 13-km grid spacing, etc.)? In other words, are these threshold values generic to be used for other model configurations/simulations?

The choice is guided by published tracking schemes with the exception of the thresholds concerning whether a constructed track is retained or discarded. This final set of thresholds is based on an analysis of the underlying data. It should be revisited if other model configurations are used. This is now stated in section 2.2.

When analyzing/comparing the tracked ACEs from the ICON-LAM ensemble member simulations, I would suggest adding the comparison against the ACEs derived from the best track data (e.g., HURDAT2) as well.

As stated in the reply to your first comment, a comparison to best track data is omitted intentionally, as it would not be very informative, given the underlying dataset.

I know the focus of this work is on storm track, intensity, and development phase (weak, strong, genesis, decaying, transition, etc.) tracking. I was wondering if the tracker used in this study can also track storm structure and size metrics (radius of maximum wind, radii of 34-kt, 50-kt, 64-kt winds, etc.).

In principle, the radius of maximum wind could be determined by the algorithm, as both the TC center and the location of maximum wind are known. However, this is currently not implemented. Radii of certain wind speeds are not as straight forward. It may best be left to an additional post-processing step for those who require these specific data.

**Reviewer #3**

This paper applies an approach of running the same tropical cyclone (TC) tracker with multiple combinations of threshold parameters across a multi-member regional ICON ensemble. The data is then "merged" allowing tracks to have an "identification percentage" -- that is, the number of combinations that flagged the particular storm as a TC. Storms are manually classified into different genesis and termination categories, with panel plot examples shown of each. The authors described potential false positives and also explore changes in accumulated metrics (e.g., accumulated cyclone energy, ACE) based on whether these false alarms (or weaker storms) are included or the allowable translation speed is changed. They generally find integrated metrics are relatively insensitive to these issues.

The idea of looking at "matched" detections across the parameter space is interesting and the work is suited to GMD since this pertains to the development of an algorithm to assess climate model performance. That said, there are many areas where I think the manuscript should be improved both scientifically and from a readability standpoint before it can be considered for publication. In particular, the model setup and data generation are poorly described and it is not 100% clear exactly what members are being analyzed and when. The work is also under-referenced, with previous papers exploring aspects such as parametric sensitivity and cyclone termination (extratropical transition) that are not cited or discussed in context with the results here. Improving both would heighten the paper's potential impact and make it more applicable for developers of numerical models and TC researchers. More comments are below.

Major comments:

How the simulations are performed and analyzed is quite unclear. Briefly, this is how I am interpreting these results. 20 ensemble members of the summer season 2013 are generated, with each ensemble member running identical LBCs (and surface forcing), except with the initial conditions shifted in time. Each ensemble member then spawns its own TCs internally which are then tracked -- that is, the simulations do not purport to be a reanalysis of the 2013 time period. Each combination of tracking parameters is used to track the TCs in each ensemble member. Only a single member is then analyzed (i.e., none of the figures seem to be showing a composite mean, but rather a snapshot from a single run) *except* for Figures 11 and 13, which include information about all 20 members. That is, the "identification percentage" reported in early figures is how many of the parameter combinations matched for a single TC snapshot in a single ensemble member.

It has been clarified at the end of section 2.1 that the intent is not to reproduce the 2013 season. The intent is to generate sets of viable TCs which can be used to validate the tracking algorithm. We did, however, add the comparison of ACE for the 2013 season to show that our simulations produce realistic seasonal TC activity.
The detection percentage shows how many individual parameter combinations found a single TC at a specific time. This is now clarified in the second paragraph of section 4.

The initial times are listed as having occurred during May 2013 but line 102 mentions June-November. Do the authors eliminate the beginning of the simulations (i.e., first 1-4 weeks) for spinup and start analysis on June 1 for all runs?
Yes, the simulations are initialized in May and evaluated from June to December. The revised manuscript now explicitly defines the season as beginning in June and ending in December, and explicitly states that the month of May is used for initialization and spin up of the simulations.

The simulations are run with ERA5 boundary conditions, do we expect TCs to be matched *across* the ensemble members or just the ACE to be correlated as in Fig. 11? Put another way, do we expect Fig. 2 to look similar if we use a different ensemble member, or are the LBCs too far away

to influence TC genesis and tracks within the center of the domain? To me, these simulations seem like a precursor to a seasonal prediction system (i.e., force with some conditions but acknowledging the model can generate different realizations of TC activity underneath these forcings) but I could be wrong.

The simulations do not reflect the observed 2013 season. This is now stated clearly in the subsection "Numerical simulations". The numerical simulations were originally performed for other purposes. In the context of this paper, they can be seen as a set of manifestations of an arbitrary season. The intent is to show that TCs are generated, and can be tracked by the algorithm.

Regarding parametric sensitivity, both Horn et al., 2014 (Tracking scheme dependence of simulated tropical cyclone response to idealized climate simulations) and Zarzycki and Ullrich, 2017 (Assessing sensitivities in algorithmic detection of tropical cyclones in climate data) are under-referenced. Both also looked at how changes in tracker settings could impact tracked TC statistics. It would be beneficial to link some of their findings regarding parameter sensitivity to those here. For example, the former paper found that threshold differences were the most important contributor to differences between modeling centers with different algorithms and the latter paper also found that integrated metrics (such as ACE) were relatively insensitive to tracker configuration. Ullrich and Zarzycki, 2017 (TempestExtremes: a framework for scale-insensitive pointwise feature tracking on unstructured grids) contains a review of previously published trackers and associated literature that may be worth exploring (e.g., adding additional references in the introduction and comparing/contrasting top-level findings in the results and discussion).

The three listed publications are now integrated into the introduction. Zarzycki and Ullrich (2017) is now also used as a reference for the results in the manuscript that assess the impact of false positives on ACE. However, this is kept rather brief, as their approach is not easily comparable to our approach. The key commonality appears to be that ACE is not very sensitive to weaker TC stages. Using geopotential layer thickness to detect a warm core (Tsutsui and Kasahara, 1996, Simulated tropical cyclones using the National Center for Atmospheric Research community climate model) has also been added to the introduction, and appendix B of Ullrich and Zarzycki (2017) is now pointed to as a review of tracking schemes.

Section 5. There is work in this space regarding using cyclone phase space to help determine genesis/lysis. Two papers that come to mind are Bieli et al., 2020 (Application of the Cyclone Phase Space to Extratropical Transition in a Global Climate Model) and Bourdin et al., 2022 (Intercomparison of Four Tropical Cyclones Detection Algorithms on ERA5). The latter is relevant to other aspects of this work (they utilize a hit rate/false alarm approach against pointwise observations to understand parametric and dataset uncertainty).

Bieli et al. (2020) and Bourdin et al. (2022) are now referenced where appropriate. In particular, the methods to treat extratropical transition are highlighted, as these seem very relevant to future developments of the algorithm. While we cannot use best tracks to assess hit rate and false alarms, parallels are drawn where appropriate.

Other comments:

Fig. 3, why does the identification percentage go down (red, bottom) when the storm intensity is going up (L to R, black curve up, blue curve down)? Are the authors discussing this when they mention "The allowance for this displacement is sensitive to the warm core threshold parameters, which is reflected in the reduction of the identification percentage for this time step. This in turn shows that the identification percentage is not sensitive to TC intensity alone"?

Yes, this sentence refers to the identification percentage being affected by the displacement of the warm core. This is now made clearer in the manuscript.

Why are the identification percentage lines not shown at the bottom of Figs. 6 and 7 as with the previous figures? I would assume these would decrease to the right with time?
The percentages are not shown because the reasons for termination are not necessarily related to the algorithm not detecting a potential TC center, so we believe that the omission of the identification percentage is appropriate.

The idea of "parallel" in the title needs to be reworded. Most people seeing that are going to interpret this as code that has been parallelized (using MPI, for example) when I believe the authors imply they are running the same code with multiple parameter combinations and then combining the results into a single track dataset.
The title has been changed to "Use of threshold parameter variation for tropical cyclone tracking."

It wouldn't hurt to add further clarity to the section describing the algorithm. For example, it appears all local minima satisfying criteria 1 are first found. Then the columns above them are scanned for vorticity. Assuming both of those checks are satisfied, the column is again checked for a T_anom maximum. All these storms are considered potential storms and then "glued" together as a second step dependent on the tau (duration) threshold.
Yes, the criteria are evaluated in sequence, which is now explicitly stated in the manuscript following the list of criteria in section 2. Further, it is now explicitly stated that the tracks are constructed in a second step.

There has been some recent work that shows that sea level pressure is better simulated in atmospheric models (Roberts et al., 2020 "Impact of Model Resolution on Tropical Cyclone Simulation Using the HighResMIP-PRIMAVERA Multimodel Ensemble") and is a better correlate to damages (Klotzbach et al., 2020 "Surface Pressure a More Skillful Predictor of Normalized Hurricane Damage than Maximum Sustained Wind"). It may be interesting to categorize storms by PS which would also eliminate the below issues regarding maximum wind and scanning radius.
We agree that categorizing TCs by surface pressure has merits. However, we show that our algorithm adequately assesses the TC activity on a seasonal scale, and believe that this is best done using ACE, as this is a very common metric used to describe seasonal activity. The implementation of the algorithm produces an output file that contains the central surface pressure as well, and there is nothing prohibiting a user from using surface pressure in any further analysis.

Lines 20-21 can be rephrased since the authors argue that manual tracking is complicated by subjectivity but then they undertake a manual tracking in part of the manuscript.
The manuscript now states that manual tracking is cumbersome and should be avoided by using an automated and objective scheme. The manual tracking that follows is reduced to the minimum required for meaningful functionality and validation of the algorithm.

Line 99. Describe R03B07 -- I assume ICON is on an unstructured grid, hence the need for remapping to a regular grid?
The manuscript now states that ICON uses an unstructured, icosahedral grid, and states where more information on the nomenclature can be found.

Line 108. Do the authors mean geopotential height? I am not sure why surface geopotential needs to be prescribed at the lateral boundary as this is commonly a constant surface boundary.
ICON can use surface geopotential as part of the boundary conditions (which is recommended in the documentation when using IFS data), which we derived from ERA5. Surface geopotential is not constant, but it has been calculated from elevation data (see https://confluence.ecmwf.int/display/CKB/ERA5%3A+data+documentation#ERA5:datadocumentation-SurfaceelevationdatasetsusedbyERA5).

Line 190. See Stern and Nolan (2012) "On the Height of the Warm Core in Tropical Cyclones" which would provide more context than the large range (most of the troposphere) described here.
A brief discussion of Stern and Nolan (2012) is now included. It provides more context for the warm core height in figure 1. Wang et al., 2019 (A 13-Year Global Climatology of Tropical Cyclone Warm-Core Structures from AIRS Data) are also cited for more information on warm core altitude.

Line 204. The 113 TCs are detected across the 20 ensemble members? So about 5-6 TCs per member?
Yes. This is now clearly stated in the manuscript.

Line 255. Why is 100 km chosen? In observations, TCs are generally O(1000 km) wide and most tracking algorithms look out at radii from 200-500 km. 100 km can be inside the RMW of even mature TCs (see annular storms). I understand the concern with picking up non-TC wind speeds, but that issue would seem to be problematic with extremely large radii, not O(250-500km).
We found that using 100 km is appropriate for the given dataset. Increasing this to 250 km or 500 km would be detrimental due to the inclusion of non-TC winds for weak systems, which are mostly affected by this. However, it is now also explicitly stated that this maximum radius should be revisited for use with other datasets. Further, weak TCs in particular are underestimated by this choice, but their impact on seasonal ACE is not large.

Line 329-331. How was this determined? If this was a manual process (as I assume it was), what factors were taken into determining a false positive? How were "edge" cases (storms that were perhaps subtropical in nature) handled, or were all the false positives as obvious as this one?
The false positives were quite clear. This is due to edge cases not being persistent, and thus filtered by the minimum lifetime criterion (which is its intended purpose). This is now stated in the opening paragraph of section 6.

Line 354-355. This would seem to be a solvable problem in that local minima could be merged or an offset between sea level pressure minimum and vorticity maximum can be allowed.
An offset between the two is already possible in the algorithm, as only a minimum vorticity value needs to be reached at the location of the sea level pressure minimum. The issue causing the falsely tracked initial location is that the actual TC center is too far displaced from the warm core, such that no warm core can be detected above the sea level pressure minimum (even though some warm core displacement is allowed by the algorithm). Further, as pointed out in the manuscript, there is only one such case, which has virtually no impact on ACE.

Line 426-430. See above comments re: r = 100km for vmax calculation.
As stated above, using 100 km seems appropriate for our data set, but should be revisited for other data.

Line 445-446. See the discussion of Bieli and Bourdin (and references therein, I assume) above, since this approach has been applied previously in tracking algorithms.
The methods in Bieli et al. (2020) and Bourdin et al. (2022) could be used for future improvements regarding the capture of the extratropical transition process. They are now both referenced accordingly.

Figures. The labels are very small and hard to read. Also, the panel titles can be cleaned up (ex: 08_ref 26 can be eliminated).
All figures have been reworked appropriately.

Line 414. Needs to be \citep{}.

\citep{} is now used.

---

## Author Response (AR2)

**Reviewer #3**

Thank you for your further review. We addressed the comments below, again in blue font.

The authors have addressed the majority of my comments. While I think there remain other interesting avenues to explore regarding both the model and tracker performance, they seem reasonable as targets for future work. I have a few remaining minor comments below.

Line 135. While the vorticity isn't calculated on a pressure level (I'd argue this is relatively trivial, but perhaps this wasn't done at runtime), an approximate pressure level can be provided to make it more comparable to previous tracker literature. For 2.5km over the ocean, I assume this is approximately 700 hPa.
The manuscript now states that this corresponds to roughly 750 hPa.

Line 147. Close parentheses.
The parentheses are now closed.

Line 155. There may be a benefit to at least reporting this first translational speed in km/day which provides a better idea of the spatial movement of a TC (and what distance normalized to 24 hours it must travel to exceed the criteria).
The manuscript now states that 20 m/s corresponds to 1728 km per day.

Line 160. Would be good to add another sentence here speculating as to why (or at least bridging with the false alarms). I assume things like warm seclusions in extratropical cyclones are flagging short-lived cyclones in the mid-latitudes.
The manuscript now names upper-level temperature gradients as the main cause for short-lived false positives, that are removed by the lifetime criterion.

Line 235. This possibility is undercut a bit by using observed SSTs (e.g., see some of the work from Malcolm Roberts et al., that showed ensembles of simulations using observed SSTs do a good job reproducing inter-annual variability (e.g., observed ACE), although I think the behavior of the model is well-within the envelope published in the TC/model literature.
Since the simulations are intended to represent arbitrary TC seasons, we believe that the wording we chose is confusing. To clarify, the manuscript now states that the simulations produce ACE values that are realistic in the sense that the values fall within a range that is reasonable for an arbitrary season, not necessarily for the 2013 season.

Line 323. Add language to specify that the track is terminated when *at least one* of the 4 things occurs.
The manuscript now states clearly that only one of the listed events need to occur for the track to be terminated.

Line 352. Since the translation cutoff is reporting in m/s, provide the translational speed across these great circle distances.
The distances and velocities are now explicitly stated.

Line 421. I assume this is the vestige of preparing a student's thesis for publication (also, see author contributions). I have no issue with using the term "thesis" but if the authors would like to change it to "manuscript" or "paper" they should feel free to.
This is indeed vestigial, and has been changed to "paper."

Figure 13. Why isn't the integral 1.0 of the bars (e.g., if you add the percentage of all the blue bars it appears to be ~0.2. Is this because they need to be also multiplied by the range (5 m/s)? Would either tweak the figure or caption to clarify.

The figure has been change to clearly show what percentage of TCs fall into a specific bin.